# Seeing Glass: Joint Point Cloud and Depth Completion for Transparent Objects

**Haoping Xu**[1,2*], **Yi Ru Wang**[1,2*], **Sagi Eppel**[1], **Alàn Aspuru-Guzik**[1,2],
**Florian Shkurti**[1,2], **Animesh Garg**[1,2,3]
[1]University of Toronto, [2]Vector Institute, [3]Nvidia
{haoping.xu, yiruhelen.wang}@mail.utoronto.ca
sagieppel@gmail.com, aspuru@utoronto.ca, {florian, garg}@cs.toronto.edu

**Abstract:** The basis of many object manipulation algorithms is RGB-D input. Yet, commodity RGB-D sensors can only provide distorted depth maps for a wide range of transparent objects due light refraction and absorption. To tackle the perception challenges posed by transparent objects, we propose TranspareNet, a joint point cloud and depth completion method, with the ability to complete the depth of transparent objects in cluttered and complex scenes, even with partially filled fluid contents within the vessels. To address the shortcomings of existing transparent object data collection schemes in literature, we also propose an automated dataset creation workflow that consists of robot-controlled image collection and vision-based automatic annotation. Through this automated workflow, we created Toronto Transparent Objects Depth Dataset (TODD), which consists of nearly 15000 RGB-D images. Our experimental evaluation demonstrates that TranspareNet outperforms existing state-of-the-art depth completion methods on multiple datasets, including Clear-Grasp, and that it also handles cluttered scenes when trained on TODD. Code and dataset will be released at https://www.pair.toronto.edu/TranspareNet/

**Keywords:** Transparent Objects, Depth Completion, 3D Perception, Dataset

## 1 Introduction

RGB-D sensors have been instrumental in 3D perception for robotics. While reliable for opaque objects, commodity-level RGB-D sensors often fail when capturing the depth of transparent objects, made of materials such as glass or clear plastic. This is because transparent objects possess unique visual properties that distort the captured depth. Specular surface reflections introduce gaps in the depth map, while depth projection to surfaces behind the transparent object, as shown in Figure 12, introduce inaccurate depth estimates. Nonetheless, transparent objects are common in our homes and daily lives, from kitchens and dining rooms to laboratory settings. Perception of these objects is under-explored, yet vital for robotic systems that operate in unstructured human-made environments.

In this work, we present a method to leverage real complex transparent object depth to estimate accurate 3D geometries of transparent objects. The design is motivated by three main ideas. First, although the depth information for transparent objects captured by RGB-D sensors is noisy by nature, there remains useful components that can be leveraged by downstream depth-completion tasks. The previous state-of-the-art method and dataset, ClearGrasp [1], masks out all depths at the location of the transparent object, renders them invalid, and conducts global optimization with surface normal and boundaries to reconstruct its geometry. This is sub-optimal, as it ignores all depth information at the transparent object's location. Ours leverages the unique depth distortion at the location of transparent objects to generate a point cloud distribution using the Point Cloud Completion module, which is then fed through a Depth Completion module to generate the complete depth map. Second, existing large-scale datasets for transparent objects are either synthetic [1], too simple and without clutter [2], or lack depth information [3]. There remains a lack of datasets for transparent objects in a real-world setting with clutter and content within. Therefore, we introduce the novel Toronto Transparent Objects Depth Dataset (TODD) which is comparable in scale to existing transparent object datasets, and captures the realistic state that transparent objects are found in everyday life,

---

*Authors contributed equally

5th Conference on Robot Learning (CoRL 2021), London, UK.

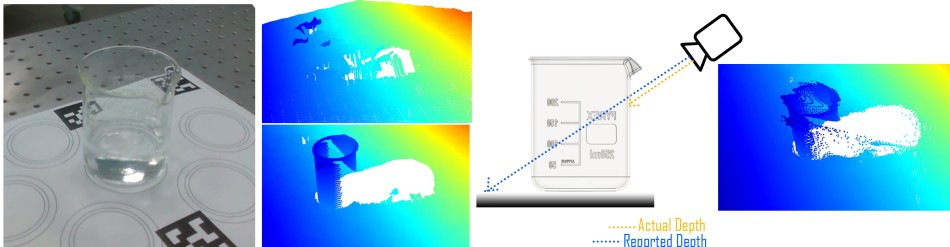

**Figure 1: Distortion in Transparent Object Depth.** From left to right, (a) RGB image, (b) Raw Depth and Ground Truth Depth, (c) Incorrect depth measurement due to reporting background depth as transparent object depth, (d) Predicted Depth from TranspareNet

including clutter, which is absent from previous datasets. Finally, collecting a dataset of transparent objects has either involved synthetic generation [1], which has a significant and often insurmountable domain gap with real world data, or manual placement and capture, which is too difficult to scale up in size. Therefore, we design an automated pipeline for dataset collection, which can capture and annotate RGB-D data, including the corrected depth for transparent objects.

Our primary contributions are threefold:

1. We introduce the Toronto Transparent Objects Depth Dataset (TODD), a large-scale real transparent object dataset with around 15,000 RGB-D observations that contain RGB, raw depths paired with ground truth depth (in which the depth at the transparent object is complete), instance segmentation, and object pose. To our knowledge, our dataset is the first large-scale transparent object dataset that contains complex scenes with clutter, and contains transparent objects in a realistic setting with fluid content within the vessels.
2. We introduce a scalable automatic dataset collection and annotation scheme for the collection of RGB-D images of transparent objects, with fully automatic labeling of ground truth depth, 6DoF pose, as well as instance segmentation.
3. We introduce a novel depth completion method, TranspareNet, that achieves state-of-the-art performance on the existing ClearGrasp dataset [1], and benchmark its performance on our dataset, TODD. TranspareNet is a joint point cloud and depth completion method that leverages RGB and depth signals of transparent objects.

## 2   Related Work

**Transparent Object Segmentation.** Transparent objects' refractive and reflective nature, as well as changing appearance in various scenes due to background, makes detecting them a challenging task in computer vision. There have been several works that applies state-of-the-art segmentation methods [4, 5, 6] to transparent object segmentation [7, 1, 8]. However, due to the significant domain gap between opaque and transparent objects, all of these aforementioned methods were trained on custom transparent object datasets. Due to the cost associated with image collection and annotation, many synthetic datasets have been proposed, including TOM-Net [9], ClearGrasp [1], and Omni [10][2]. While real world transparent object datasets, like Trans10K [8] and LabPic [7], avoid the potential domain gap between rendered and real images, they are time-consuming and expensive to collect and annotate. LIT [11] utilizes a special light-field sensor to detect transparent objects and estimate their poses. Our proposed automated dataset collection pipeline enables large scale, automatic annotation of transparent objects using common RGB-D sensor.

**Depth Completion.** Depth completion refers to the task of generating a dense depth map from an incomplete depth map due to sparse measurements, noise, or sensor limitations. In the context of our work, we use depth completion to describe the task of completing and refining noisy depths captured by RGB-D sensors for transparent objects. Works in this area generally fall in four main schemes: constraints driven, monocular depth estimation, depth completion from sparse point cloud, and depth completion given noisy RGB-D. Our work falls in the last scheme. Constraint driven approaches assume a specific setup method with fixed viewpoint(s) and capturing procedure [12, 13, 14, 15, 16], sensor type [17, 18, 11], or known object models [19, 20, 21]. Our proposed depth completion

---

[2]Concurrent work, dataset is not released

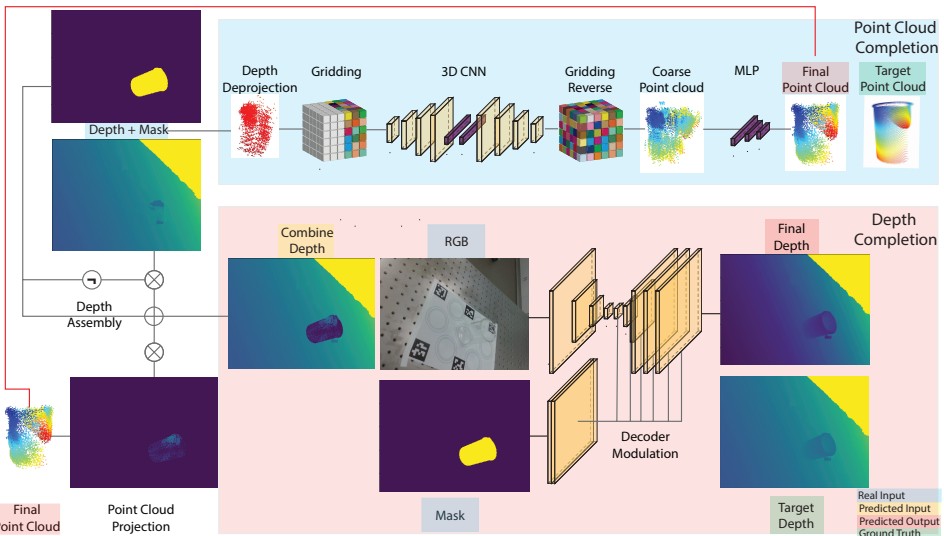

**Figure 2: Overview of the proposed TranspareNet.** Transparent object depth is de-projected to a point cloud, and put through Point Cloud Completion module to get the final point cloud. The final point cloud is then projected to the depth domain and replaces the original depth of the transparent object within the mask. An encoder-decoder based Depth Completion module takes combined depth and RGB signal as input, and the decoder modulation branch takes the object mask as input for modulation. Finally, the decoder outputs the predicted completed depth.

method does not apply any assumptions. Monocular depth estimation refers to the direct regression of depth from RGB input [22, 23, 24, 25, 26, 27, 28, 29, 30, 31, 32, 33, 34, 35]. This family of works generally require access to large-scale RGB-D datasets, for which the depth reflects objects depicted in the RGB image. Historically, this area has not been explored for transparent object handling, due to the absence of large scale transparent data paired with noise-free depth. Our proposed dataset and data collection and annotation approach can now facilitate the generation of sufficient data to support research in this direction. The third scheme, depth completion from sparse depth data, refers to the conversion of a sparse point cloud to a dense depth map via deep learning methods. Works in this area concentrate on applications involving LiDAR for outdoor environments [36, 37, 38, 39], and are inherently different from the semi-sparse and noisy signals from RGB-D sensors for transparent objects. There are two main lines of work in depth refinement of noisy RGB-D data of transparent objects. ClearGrasp [1] takes an optimization approach and estimates surface normals, occlusion boundary, and segmentation to perform a global optimization for depth estimation. More recently, [10] proposed a voxel based local implicit neural function for depth estimation. We show that our method outperforms both of these methods. The key insight that contributed to the success of our method is that we leverage the unique distortion in the depth caused by transparency to generate a coarse estimation of object depth through point cloud completion, and conduct depth completion for refinement of the sparse point cloud. Previous methods like ClearGrasp [1] discard this information.

**Prior Transparent Dataset Collection and Annotation**. Trans10K[8] is a 2D transparent object segmentation dataset that consists of real images of transparent objects and their semantic segmentation masks. All of its images and labels are captured and annotated manually. Schenck and Fox [40] proposed a dataset of transparent liquid in opaque containers, and consists of rendered and real-world images. The captured images' segmentation masks are annotated via thermal cameras in combination with heated liquid. For transparent object datasets with depth information, ClearGrasp [1] uses a synthetic dataset generated by Blender as the training set, and collects the small scale real-world validation and testing dataset using matched transparent and opaque objects. This process requires manual placement and matching of transparent and opaque pairs for each generated image. Another 3D dataset, Keypose [2], collects its dataset using a eye-in-hand robot arm and tracking system based on AprilTag 2 [41]. The camera path is calculated using AprilTag, and along the scan trajectory, several frames' keypoints are manually labelled, where the rest are labeled based on the trajectory. The depth info is collected using a similar replacing method as ClearGrasp [1]. Such replacing method needs human intervention to align the opaque and transparent twins through image overlay, which can be hard and inaccurate especially in complex scenes. Furthermore, since each image in ClearGrasp [1] is manually captured, opaque swapped and annotated, it cannot scale to a

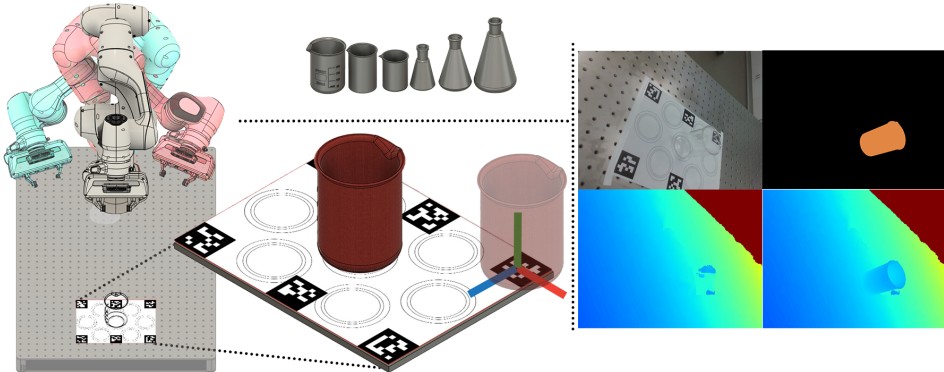

**Figure 3: Dataset creation pipeline.** A commodity RGB-D sensor is mounted to the robot arm's end effector. The scene with the transparent object is scanned from multiple viewing angles to collect the raw depth (left). AprilTags on the base template are detected for each image, and based on their 6DoF poses and the known translation between tags and objects, we can fit the 3D model of object(s) to their respective locations (middle). The result of this automatic collection and annotation process is the RGB image, instance object segmentation, raw depth, ground truth depth (right).

larger dataset size due to time and labour constraints. Our method overcomes the manual effort that previous methods used for dataset collection with an automated pipeline for dataset creation and annotation.

## 3 Toronto Transparent Objects Depth Dataset (TODD)

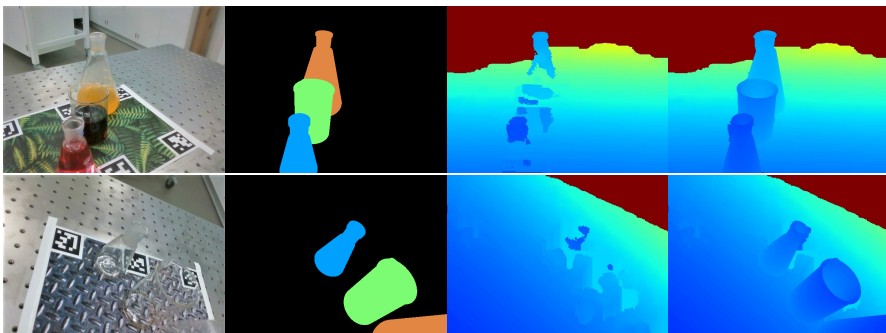

**Figure 4: Samples from the proposed dataset.** (a) RBG image (b) Instance Segmentation (c) Raw depth from RGB-D sensor (d) Ground truth depth obtained through automatic depth annotation. The dataset also includes 6DoF object pose information, which is not depicted in the figure.

We use an eye-in-hand Franka Emika Panda controlled by FrankaPy[42] with Intel Realsense RGB-D camera to collect the dataset. As shown in Figure 3, the pipeline consists of multiple steps. First, the end-effector of the robot arm moves the camera to multiple positions around the transparent object(s), while ensuring focus on the transparent object. Templates with AprilTags [41] and alignment marks are used to maintain the known translation between objects and tags. Then for every viewpoint captured, AprilTags can be recognized together with their 6DoF pose in camera coordinates. Combining this with the known shift between tags and transparent objects, the corresponding CAD model of each object can be overlaid at the appropriate locations of transparent objects within the image. The resultant 3D meshes are sufficient to automate the succeeding dataset annotation tasks. Namely, projection of meshes to the image space provides the instance segmentation mask as well as the ground truth depth as shown in Figure 4. With our proposed dataset creation pipeline, we can collect and annotate 300 RGB-D images for one scene in 30 minutes with minimal human intervention. Compared to methods used by [1, 3, 2], our proposed pipeline only needs placing the transparent object once per sequence and annotation is fully automated, which guarantees the accuracy of annotated labels.

In total, TODD has 14,659 images of scenes which contains six glass beakers and flasks in five different backgrounds. Four objects are used in training set and the other two novel objects form

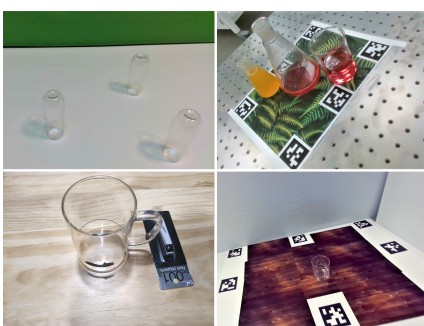

**Figure 5: Sample images** of ClearGrasp [1] (top left), Ours (top right), Trans10K [3] (bottom left) and KeyPose [2] (bottom right).

|  | ClearGrasp [1] | Trans10K [3] | KeyPose [2] | **Ours** |
|---|---|---|---|---|
| Real Samples | $0.3 \times 10^3$ | $10 \times 10^3$ | $40 \times 10^3$ | $15 \times 10^3$ |
| Auto Collection & Annotation | ✗ | ✗ | ✗ | ✔ |
| Raw & Ground Truth Depth | ✔ | ✗ | ✔ | ✔ |
| Instance Segmentation | ✗ | ✗ | ✔ | ✔ |
| RGB | ✔ | ✔ | ✔ | ✔ |
| Pose | ✗ | ✗ | ✔ | ✔ |
| Multi-Object Clutter | ✔ | ✔ | ✗ | ✔ |

**Table 1:** Comparison of TODD (Ours) with ClearGrasp [1], Trans10K [3], Keypose [2]. Ours is the only large-scale automatically collected 3D transparent object dataset with real glassware in cluttered settings.

the novel validation and test set. The training set has 10,302 images where validation and testing set combined has 4357 images. When comparing with existing datasets in Figure 5, TODD has the following advantages. Every scene consists of up to three transparent objects with occlusion, which introduces additional complexity to the dataset compared to KeyPose's [2] single object scenes. The objects and their placement are selected to mimic real-life transparent glassware, which can help to develop vision aware robots capable of manipulating transparent vessels. One particular direction of that is robotic chemists automating chemical research [43]. Additionally, the glass vessels in the dataset are empty or filled with 5 different coloured liquids to simulate real-life circumstances, which is common in 2D transparent datasets like Trans10K [8] and LabPic [7], yet is missing in 3D datasets like ClearGrasp [1] and KeyPose [2]. Besides the RGB and raw depth information, the dataset consists of ground truth depth, instance segmentation mask, as well as the objects' 6DoF poses. Our dataset's size is much bigger than ClearGrasp's [1] 286 real world images, and is comparable with Trans10K (10K) [8] and KeyPose(48K)[2], while having the potential to quickly scale to greater size with relative ease.

## 4 Our Depth Completion Method

Existing transparent object depth completion and pose estimation methods consider the distorted and incomplete depth of transparent objects as misguiding information and either only rely on color information [2], or crop out the incorrect depth [1]. We propose a novel depth completion method to handle transparent objects called TranspareNet, whose general pipeline is shown in Figure 2. Instead of discarding the transparent objects' depth, each item's depth is separated out and de-projected into a point cloud. With the distorted point cloud, a Point Cloud Completion network is used to recover the basic shape in the form of a predicted point cloud. However, such a point cloud is still too sparse and noisy to be considered as the final complete depth and used in downstream manipulation tasks. To further refine the depth, predicted point clouds are projected back to the depth channel and a depth completion module is used to fill the blank depth as well as correct the shifted depth. By combining both point cloud and depth completion tasks, our method avoids directly feeding incorrect transparent depth to the depth completion task, while capable to utilize the unique distortion depth caused by transparency and generate a coarse estimation of object depth via point cloud completion. Such intermediate depth improves the performance of the downstream depth completion subtask, as shown in Section 5.

### 4.1 Point Cloud Completion

Transparent objects cause the RGB-D sensor to report invalid or inaccurate depth information. The Point Cloud Completion module is aimed at taking each object's depth as point cloud and estimating the correct depth by predicting the completed point cloud. Inspired by the point cloud processing modules proposed in GRNet [44], we first feed the incomplete point cloud from transparent object depth de-projection through a Gridding layer [44], which computes weighted vertices of 3D grid cells that points lie in. Then we use a 3D CNN encoder-decoder with U-Net connections [45] to learn the features that are necessary for point cloud completion. The succeeding component is the Gridding reverse layer [44], which back projects each 3D grid cell to a point in the coarse complete

**Table 2: Depth completion results** on Known and Novel Objects within the ClearGrasp [1] Dataset. Metrics are defined in Section 5.1. Arrows beside the metrics denote whether higher or lower values are desired.

| | RMSE ($\downarrow$) | MAE ($\downarrow$) | $\delta_{1.05}$ ($\uparrow$) | $\delta_{1.10}$ ($\uparrow$) | $\delta_{1.25}$ ($\uparrow$) | REL ($\downarrow$) |
|---|---|---|---|---|---|---|
| Seen | | | | | | |
| NLSPN [48] | 0.136 | 0.113 | 0.1902 | 0.3595 | 0.7043 | 0.231 |
| ClearGrasp [1] | 0.041 | 0.031 | 0.6943 | 0.8917 | 0.9674 | 0.055 |
| LocalImplicit [10] | 0.023 | 0.017 | 0.8356 | 0.9504 | 0.9923 | 0.031 |
| TranspareNet-DC Only | **0.011** | **0.008** | **0.9354** | **0.9831** | **0.9985** | **0.010** |
| Unseen | | | | | | |
| NLSPN [48] | 0.132 | 0.106 | 0.1625 | 0.3213 | 0.6478 | 0.239 |
| ClearGrasp [1] | 0.044 | 0.038 | 0.4137 | 0.7920 | 0.9729 | 0.074 |
| LocalImplicit [10] | 0.041 | 0.034 | 0.5269 | 0.7942 | 0.9805 | 0.063 |
| TranspareNet-DC Only | **0.032** | **0.027** | **0.6080** | **0.8053** | **0.9821** | **0.052** |

cloud whose coordinate is the weighted sum of the eight vertices. This is then forward propagated through a Multi-Layer Perceptron that creates the final completed point cloud. For each point, its corresponding grid cell's features from the CNN decoder are concatenated and combined with it. The MLP also refines the coarse to final point cloud with the aim of recovering details of the target object. However, the 3D CNN based network's limited resolution leads to sparse and noisy prediction, thus an additional depth completion module is needed.

### 4.2 Depth Completion

The output from the depth completion module consists of a sparse distribution around the location of the transparent vessel(s) shown in Figure 2. We fuse this with the RGB signal to generate a 4D input to our encoder-decoder structure. The encoder-decoder based depth-completion module processes the scene's depth to fill in the sparse depth and correct the noisy depth present. The decoder of the Depth Completion module consists of spatially-adaptive denormalization (SPADE) blocks, first introduced in [46]. Our usage of SPADE in the encoder-decoder Depth Completion module is a variant of [47]. This module enables us to learn spatially-dependent scale and bias for decoder feature maps, which helps reduce the domain shift between RGB and depth, as introduced by the empty depths on the depth map. Let $\mathbf{m}$ denote the object mask. Given a batch of N inputs with dimension $C^i \times H^i \times W^i$, the output of the SPADE block at site ($n \in N$, $c \in C^i$, $y \in H^i$, $x \in W^i$) is then

$$\gamma^i_{c,y,x}(\mathbf{m}) = \frac{h^i_{n,c,y,x} - \mu^i_c}{\sigma^i_c} + \beta^i_{c,y,x}(\mathbf{m}) \quad \text{where } \mu^i_c = \frac{\sum_{n,y,x} h^i_{n,c,y,x}}{NH^iW^i} \quad \sigma^i_c = \sqrt{\frac{\sum_{n,y,x} (h^i_{n,c,y,x})^2 - (\mu^i_c)^2}{NH^iW^i}} \quad (1)$$

$h^i_{n,c,y,x}$ is the activation at the site prior to normalization, $\mu^i_c$ and $\sigma^i_c$ are the mean and standard deviation of the activations within channel $c$. $\gamma^i_{c,y,x}(\mathbf{m})$ and $\beta^i_{c,y,x}(\mathbf{m})$ represent the modulation parameters. They represent the scaling and bias values for the $i$-th activation map at site $(c, y, x)$, respectively.

### 4.3 Loss Function

The point cloud completion network is trained with Gridding Loss [44], which is a L1 distance between predicted $\mathcal{G}_p = < V^p, W^p >$ and ground truth $\mathcal{G}_{gt} = < V^{gt}, W^{gt} >$ 3D grids in $N_G$ resolution. $V = \{v_i\}_{i=1}^{N_G^3}$ is collection of all vertices in 3D grid and $W = \{w_i\}_{i=1}^{N_G^3}$ is the weights corresponding to each vertex. Gridding Loss bypasses the un-orderedness of point clouds and is evaluated on the 3D grid. The depth completion network is trained using log $L_1$ pair-wise loss which forces the pairs of pixels in the predicted depth to regress to similar values as the corresponding pairs in the ground truth depth [47]. Let $\mathcal{G}$ describe the set of pixels where the ground truth depth is non-zero, $i$ and $j$ are the pixel pairs, and $y$ and $y*$ denote the ground truth and predicted depths, respectively. We express these two loss functions as:

$$\text{Gridding:} \mathcal{L}(W^p, W^{gt}) = \frac{1}{N_G^3} \sum |W^p - W^{gt}|, \quad \log L_1: \mathcal{L}(y_i, y_i^*) = \frac{1}{|\mathcal{G}^2|} \sum_{i,j \in \mathcal{G}} |\log \frac{y_i}{y_j} - \log \frac{y_i^*}{y_j^*}| \quad (2)$$

## 5 Experimental Results

We conduct experiments to test our TranspareNet model along with SOTA methods like NLSPN [48] and ClearGrasp[1] as well as superior con-current method LocalImplicit [10] on the ClearGrasp

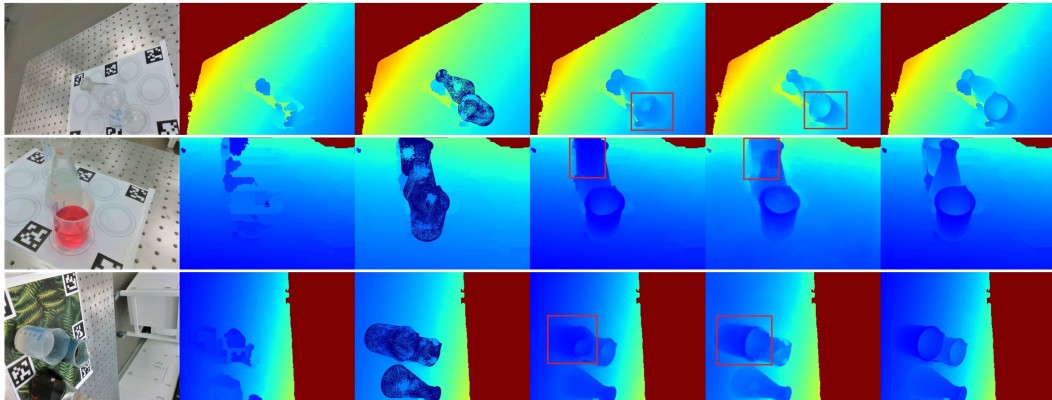

**Figure 6: Visualization of prediction results on TODD dataset.** From left to right, (a) RGB image (b) Raw depth from RGB-D sensor (c) PCC predicted depth (d) DC predicted depth (e) TranspareNet predicted depth (f) Ground Truth. We mark the major difference between DC and TranspareNet predictions with bounding boxes.

[1] dataset to demonstrate our method's strong performance. Additionally, we test ClearGrasp, TranspareNet, as well as its point cloud and depth completion modules on our TODD dataset. This allows us to study the effects of each individual module of TranspareNet and advantages of the proposed joint pipeline over existing methods for depth and point cloud completion. Importantly, our results demonstrate that distorted transparent object depth can be converted into sparse depth estimate via point cloud completion and used to improve downstream depth completion quality.

## 5.1 Metrics

For depth completion, the standard metrics as described in [1] are followed. The prediction and ground truth arrays are first resized to $144 \times 256$ resolution prior to evaluation. Errors are computed using the following metrics, Root Mean Squared Error (RMSE), Absolute Relative Difference (REL), Mean Absolute Error (MAE) and Threshold ($\delta$).

$$\text{RMSE: } \sqrt{\frac{1}{|\hat{D}|}\sum_{d_i \in \hat{D}} ||d_i - d_i^*||^2} \qquad \text{REL: } \frac{1}{|\hat{D}|}\sum_{d_i \in \hat{D}} |d_i - d_i^*|/d_i^* \qquad \text{MAE: } \frac{1}{|\hat{D}|}\sum_{d_i \in \hat{D}} |d_i - d_i^*| \qquad (3)$$

Threshold is % of $d_i \in \hat{D}$ satisfying $\max(\frac{d_i}{d_i^*}, \frac{d_i^*}{d_i}) < \delta$. Here, $\hat{D}$ denotes the set of pixels $\mathcal{G} \cap D$, which has a valid corresponding ground-truth depth, and falls within the mask. $d_i \in \hat{D}$ is the predicted depth, and $d_i^*$ is the corresponding ground-truth depth. Note that RMSE, REL and MAE metrics are computed in meters. For the threshold metric, $\delta$ is set to 1.05, 1.10 and 1.25.

## 5.2 Depth Completion Comparison with State-of-the-Art Methods

The ability of our method to estimate depth for transparent objects is shown Table 2. All models were trained using the ClearGrasp [1] dataset. *Seen* denotes synthetic objects which have appeared in training set, under a different setting. *Unseen* denotes synthetic objects that have not appeared in the training set. We compare with several state-of-the-art methods. NLSPN [48] is a state-of-the-art method for depth completion on the NYUV2 [49] and KITTI [37] datasets. ClearGrasp [1] is the state-of-the art of transparent object depth completion using global optimization. LocalImplicit [10] is a new con-current method that achieved superior performance than ClearGrasp under several metrics. The Point Cloud Completion (PCC) module cannot be trained using the ClearGrasp synthetic dataset, since it lacks raw sensor depth. Therefore, we only show the performance of the Depth Completion (DC) component of our proposed method and demonstrate that it already outperforms all depth completion methods for the metrics presented, by a large margin.

## 5.3 Performance on Our Dataset

We evaluate each component of our model on our proposed dataset as well as compare the results with performance of the ClearGrasp algorithm [1]. Evaluation of ClearGrasp uses the pre-pretrained model, since we cannot re-train ClearGrasp [1] on our data, as ground truth surface normal and

**Table 3: Depth completion results on the TODD Dataset.** We assess the performance of various models with Novel 1 Object images, as well as novel cluttered images with 2 or 3 objects. Our method outperforms all previous methods. Metrics are defined in Section 5.1. The arrows beside the metrics denote whether lower or higher values are more desired.

| | RMSE ($\downarrow$) | MAE ($\downarrow$) | $\delta_{1.05}$ ($\uparrow$) | $\delta_{1.10}$ ($\uparrow$) | $\delta_{1.25}$ ($\uparrow$) | REL ($\downarrow$) |
|---|---|---|---|---|---|---|
| | | | Novel 1 Object Scene | | | |
| ClearGrasp [1] | 0.0425 | 0.0367 | 0.4023 | 0.6018 | 0.8990 | 0.1073 |
| TranspareNet-PCC Only | 0.2443 | 0.1713 | 0.4927 | 0.5113 | 0.5196 | 0.4877 |
| TranspareNet-DC Only | 0.0188 | 0.0159 | 0.6483 | 0.8819 | 0.9931 | 0.0473 |
| TranspareNet (ours) | **0.0166** | **0.0140** | **0.7133** | **0.9299** | **0.9945** | **0.0398** |
| | | | Novel 2 Object Scene | | | |
| ClearGrasp [1] | 0.0534 | 0.0433 | 0.3458 | 0.5414 | 0.8412 | 0.1455 |
| TranspareNet-PCC Only | 0.2477 | 0.1817 | 0.4139 | 0.4434 | 0.4561 | 0.5516 |
| TranspareNet-DC Only | 0.0212 | 0.0168 | 0.5954 | 0.8256 | 0.9874 | 0.0564 |
| TranspareNet (ours) | **0.0194** | **0.0159** | **0.6475** | **0.8693** | **0.9876** | **0.0496** |
| | | | Novel 3 Object Scene | | | |
| ClearGrasp [1] | 0.0612 | 0.0493 | 0.2985 | 0.4954 | 0.8361 | 0.1536 |
| TranspareNet-PCC Only | 0.2659 | 0.1922 | 0.4275 | 0.4564 | 0.4724 | 0.5362 |
| TranspareNet-DC Only | 0.0250 | **0.0189** | **0.5902** | 0.8305 | 0.9866 | 0.0555 |
| TranspareNet (ours) | **0.0232** | 0.0190 | 0.5817 | **0.8408** | **0.9904** | **0.0546** |
| | | | Novel Combined | | | |
| ClearGrasp [1] | 0.0563 | 0.0455 | 0.3262 | 0.5233 | 0.8476 | 0.1435 |
| TranspareNet-PCC Only | 0.2584 | 0.1864 | 0.4354 | 0.4627 | 0.4767 | 0.5314 |
| TranspareNet-DC Only | 0.0232 | 0.0180 | 0.6010 | 0.8380 | 0.9879 | 0.0543 |
| TranspareNet (ours) | **0.0213** | **0.0175** | **0.6180** | **0.8619** | **0.9905** | **0.0510** |

occlusion boundaries are non-trivial to obtain for our real-object dataset TODD. For evaluation of the pretrained ClearGrasp model, we use the ground truth depth and mask [3]. When we evaluate the direct output from our Point Cloud Completion (PCC) module, we see that it suffers due to the sparsity of depths in the transparent object region, empty depth values of which are interpreted as zero to ensure numerical stability. When computing metrics using only regions within the object mask where the depth is non-empty, we see improved metric values (See Appendix), this means that the Point Cloud Completion produces meaningful depths that is valuable for the downstream depth completion task. We cannot directly use the output from the Point Cloud Completion module, since the point cloud is too sparse. When only the depth completion (DC) module is used without point cloud completion, we see inferior results as compared to our joint point cloud and depth completion network, TranspareNet, shown in Table 3. We can see that TranspareNet outperforms ClearGrasp [1], PCC, and DC. This means that our joint approach, which involves leveraging the unique depth distortion around transparent objects, generating a course estimation, and conducting depth completion using the point cloud distribution is indeed effective. A qualitative comparison of different stages of our model is shown in Figure 6. We see that for single and multi-object cluttered scenes, our method achieves SOTA results.

## 6 Conclusion

We introduced TranspareNet, a novel method that achieves state-of-the-art performance on synthetic transparent object data from ClearGrasp [1]. Our method leverages existing depth information at the location of transparent objects to estimate the complete depth for transparent objects. We also introduced a novel real dataset, TODD, which has complex scenes with numerous vessels under occlusion. The transparent vessels in our dataset mimics vessels found in household settings in shape and appearance. To our knowledge, our dataset is the first to contain depth information of transparent vessels with partially filled liquids. We benchmark our dataset using the proposed depth completion method, TranspareNet, and achieve SOTA results. We also introduced an automatic data capture and annotation workflow that consists of robot-controlled image collection and vision-based automatic annotation. We hope that this work can accelerate future research in the field of household robotics and handling of glassware.

---

[3]We acknowledge the transparent object depth completion method [10], which is concurrent with our work. However, because neither the code nor their dataset is released, we can not apply our dataset to evaluate the model's performance.

**Acknowledgments**

Animesh Garg is a CIFAR AI Chair. Alàn Aspuru-Guzik is a CIFAR AI Chair and CIFAR Lebovic Fellow. Animesh Garg and Florian Shkurti are also supported in part through the NSERC Discovery Grants Program. The authors would like to acknowledge Vector Institute and ComputeCanada for computing services. Alàn Aspuru-Guzik and Haoping Xu thank the Canada 150 Research Chair funding from NSERC, Canada. Alàn Aspuru-Guzik is thankful for the generous support of Dr. Anders G. Frøseth. The authors would like to thank Yuchi Zhao for constructive feedback and discussions on the manuscript.

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
