# OpenReview forum: "Seeing Glass: Joint Point-Cloud and Depth Completion for Transparent Objects"
_robot-learning.org/CoRL/2021/Conference — CoRL2021 Oral_

### Official Review · Reviewer_vich · 2021-07-20

**Originality:** Very Good
**Technical Quality:** Good
**Clarity Of Presentation:** Very Good
**Impact:** 4

**Recommendation:**

Strong Accept: I recommend accepting the paper and will argue for my recommendation even if other reviewers hold a different opinion.

**Summary:**

The paper presents a novel method of monocular depth completion for transparent objects. It introduces a novel architecture to be able to correct and fill in the gaps made by a typical RGB-D sensor. In order to achieve that they first mask the depth input and transform it to a point cloud, which is then feed to a 3D-CNN architecture. After that the result is projected back into the depth image and handed with the RGB image to a U-Net architecture to combine the features of both to create a complete depth reconstruction of the transparent object. Additionally, they provide a novel dataset called Transparent Object Depth Dataset (TODD), which contains 14,659 RGB-D images of different backgrounds, which was created via an automatized process, which makes it easier to create more data points in the future.

**Issues:**

* As mentioned above, it must be clear how the mask is produced and where it would come from in a real scenario
* Figure 2 must be reorganized to make it easier for a reader to understand the approach
* All figures **must** be explained in the text, this is not optional.
* All mentioned variables **must** be explained or must be referenced if they are used in a similar fashion to a prior work. In particular in section 4.2 and 4.3

**Reviewer Expertise:**

Very good: Comprehensive knowledge of the area

**Strengths And Weaknesses:**

Strengths:
* a novel architecture combining a point cloud filling approach with an image based depth completion method to tackle the difficult task of transparent object depth completion
* a new dataset, which will be helpful in the future evaluation of methods in this area
* an automatized method to generate the said dataset
* a nice in depth comparison to ClearGrasp [1]
* well written and most of the core concepts are well explained

Weaknesses:
* The biggest weakness I see is that the authors fail to mention, where the mask in Figure 2 comes from. It is not predicted as the edges of the mask look to perfect, but for a real world application it seems weird that the input requires a mask of the transparent object, as this is likely to not be available. Of course a second method could have been trained on that, but that is not mentioned in the paper, which means that in the current version, this can not be used directly on a robot.
* Sadly, the authors do not mention how the automated dataset construction affects the performance of the network. As the april tags are visible in all RGB images and the objects are always on seven fixed positions in relation to the April tags, one could ask themselves if the network is just really good in locating the april tags and then therefore predicting objects on these seven positions? I would have loved to seen, in an ablation study what happens if those april tags are removed from the scene, to ensure that the network didn't just learn the relative positions to the april tags. Especially, as the amount of training/tested objects is quite low, so that the network could have been able to recognize the category and the given april tag position, which is a much easier task then reconstructing a depth image.
* Figure 2 is hard to read, the upper and lower part should be visually separated, maybe by putting them into boxes. Additionally, why are the input images rotated in the top left corner, that makes it so much harder to understand. Furthermore, is not intuitive that the output from the top right corner is used in the bottom left corner. That should be visualized better.
* Not all figures are explained in the text, which is in my view a must. The paper contains six figures, but only 1, 2 and 6 are mentioned in the text. Figure 3, which is extremely crucial to understand how the objects are placed and 6D pose estimated is not explained in the text or even mentioned.  Figure 4 and 5 should also be mentioned.
* Section 4.2 and 4.3 introduce a lot of variables, which are not all explained in the text. If they are the same as in [46, 47] then this should be mentioned. In particular in 4.3 $V^p, W^p$ are mentioned but not explained. $W^i$ is used before, but there it seems to be the width of the image.

**Summary Of Recommendation:**

I usually don't give weak responses to papers. But, I truly believe that this paper deserve a strong accept as the idea and the presentation is good. It just lacks a few points, which are sadly to important to be overlooked. The paper sadly omits the detail, where the mask comes from, which is used in Figure 2 as an input to the method. Without that mask the whole approach does not work and therefore an application in a real scenario is out of reach, as this conference highlights the importance of applicable robotic work.

I hope the authors can clarify the usage of this mask as an input and how they plan to produce it in a real scenario, not just in the constructed dataset test. If they can show that it is also possible to train a mask detection network and use it as an input, with similar or even worse results, than I will change my review to a strong accept.

After reading the perfect response from the authors, I change my recommendation to "Strong Accept". They were able to provide satisfactory results to all my questions.

---

> ### Author Response · Authors · 2021-08-25
> **Mask Segmentation Network & Ablation Study with April Tags Removed**
>
> We thank the reviewer for the in depth review of our work, as well as the constructive feedback that was provided. We respond to the points raised as follows:
>
> **P1: “… where the mask in Figure 2 comes from. It is not predicted as the edges of the mask look too perfect, but for a real world application it seems weird that the input requires a mask of the transparent object, as this is likely to not be available.” and “If they can show that it is also possible to train a mask detection network and use it as an input, with similar or even worse results, then I will change my review to a strong accept.”**
>
> A1: Thank you for pointing this out. Training of the network requires an input mask of the transparent objects. To train the point cloud completion component of the network, we originally used the ground truth segmentation mask. For the dense depth completion component, we use the output of the point cloud completion, and generate a mask for regions where the depth is empty. We performed additional experiments, and we show below that a ground truth segmentation mask for point cloud completion training is not at all a requirement, and that our depth completion performance is maintained, even when we use a network that predicts the segmentation mask.
>
> There are several lines of work which directly target transparent object segmentation given an RGB sensor input, which we mentioned in Related Works - Transparent Object Segmentation. While the presented results used the ground truth mask, we additionally provide the results which use the Trans10k [15] segmented mask in Appendix A.7. Using the pre-trained Trans10k model for mask inference, the quality of the inference is 0.48 IoU, as mentioned in Appendix A.7. We then use the Trans10k predicted masks as input to our network, along with the raw depth, and find the quantitative results as provided in Table 4 of the Appendix. We provide the table here for quick reference as well:
>
> |                      | RMSE   | MAE    | $\delta$ 1.05 | $\delta$ 1.10 | $\delta$ 1.25 | REL    |
> |----------------------|--------|--------|------------|------------|------------|--------|
> | TranspareNet-DC Only | 0.0446 | 0.0362 | 0.4064     | 0.6056     | 0.8511     | 0.0914 |
> | TranspareNet         | 0.0341 | 0.0279 | 0.4740     | 0.7056     | 0.9102     | 0.0598 |
>
>
> We also provide representative examples  of the predicted depths using predicted masks, in comparison to predicted depths using ground truth masks in Figure 8 of the Appendix. We anticipate that with more refined masks (generated by retraining/finetuning the Trans10k model on our dataset), the results can be further improved.
>
> We believe this shows the applicability of our model in a real world scenario, and that it is indeed possible and practically useful to train a mask detection network and use the generated masks as input to our network.
>
> **P2: I would have loved to seen, in an ablation study what happens if those april tags are removed from the scene, to ensure that the network didn't just learn the relative positions to the april tags.**
>
> We provide an additional ablation study in the Appendix A.8, where we generated a mask of the April Tags present in the scene, and masked it out with gray pixels in the RGB stream. We provide a comparison of the inference results using TranspareNet with and without the removal of tags in the RGB stream in Table 5. We include a copy of the table here for reference:
>
> |                      |  RMSE  | MAE    | $\delta$ 1.05 | $\delta$ 1.10 | $\delta$ 1.25 | REL    |
> |----------------------|:------:|--------|---------------|---------------|---------------|--------|
> | **Novel 1 Object Scene** |
> |Tags Removed |0.0260 |0.0230 |0.3735 |0.7451 |0.9855 |0.0614|
> |Tags Not Removed |0.0166 |0.0140 |0.7133 |0.9299 |0.9945 |0.0398|
> | **Novel 2 Object Scene** |
> |Tags Removed |0.0288 |0.0233 |0.3611 |0.6649 |0.9665 |0.0687 |
> |Tags Not Removed |0.0194 | 0.0159 |0.6475 |0.8693 |0.9876 |0.0496|
> | **Novel 3 Object Scene** |
> |Tags Removed |0.0329 |0.0282 |0.3189 | 0.6332 |0.9704 | 0.0739|
> |Tags Not Removed |0.0232 |0.0190 |0.5817 |0.8408 |0.9904 |0.0546|
> | **Novel Combined** |
> |Tags Removed | 0.0309 |0.0271 |0.3293 |0.6452 |0.9706 |0.0707|
> |Tags Not Removed |0.0213 |0.0175 |0.6180 |0.8619 |0.9905 |0.0510|
>
>
> We also provide a qualitative analysis of the depth estimations for TranspareNet, when the April Tags are masked out in the RGB input in Figure 9 of Appendix.
>
> As can be seen, with and without the presence of April Tags, the model is able to generate very similar depth estimations. This shows that the network indeed learned the geometry of the glass vessels, and not just the relative positions to the april tags.

---

> > ### Comment · Reviewer_vich · 2021-08-31
> > **Perfect response**
> >
> > First of all thanks for this detailed answer to my questions, this could be a template on how to respond to reviews.
> >
> > I am now convinced that this paper makes a great addition to CORL and will change my recommendation to a "Strong accept".
> >
> > One last comment, I am intrigued why the performance goes down a bit when removing the April tags. I already suspected that, of course analyzing this is far beyond the scope of this paper, nonetheless is it intriguing.

---

> ### Author Response · Authors · 2021-08-25
> **Revised Figure 2 & Figure + Variable Explanations**
>
> **P3: Figure 2 is hard to read.**
>
> Thank you for pointing this out. We have updated the submission with a modified Figure 2. Changes include:
> * Visually separating the upper and lower modules
> * Making the input images non-rotated
> * Adding a connection from the output of the upper module to the input of the lower module.
>
>
> **P4: Not all figures are explained in the text, which is in my view a must.**
>
> We have added explanations for all figures at the following line numbers: figure 3 in line 120, figure 4 in line 129 and figure 5 in line 137.
>
> **P5: Section 4.2 and 4.3 introduce a lot of variables, which are not all explained in the text.**
>
> We have updated the submission with explanations of the notations used.
>
> In line 190, we provide an explanation for the notations used for Equation 1.
> > “ Let $\textbf{m}$ denote the object mask. Given a batch of N inputs with dimension $C^i \times H^i \times W^i$, the output of the SPADE block at site ($n \in N$, $c \in C^i$, $y \in H^i$, $x \in W^i$)“
>
> In line 199, we provide an explanation for the notations used for Equation 2.
> > “$V$={$v_i$} is collection of all vertices in 3D grid and $W$={$w_i$} is the weights corresponding to each vertex.”

---

> ### Author Response · Authors · 2021-08-30
> **Thanks for your time, any additional feedback is welcome!**
>
> Dear Reviewer vich,
>
> We have addressed *all* the points that you raised with additional clarifications and experiments. Please take a look at our comments, the revised paper and supplemental materials (the latexdiff versions with our changes highlighted are included in the supplementary material for reference). We'd appreciate it if you could let us know if there are any remaining concerns for you.

---

### Official Review · Reviewer_UETv · 2021-07-21

**Originality:** Very Good
**Technical Quality:** Very Good
**Clarity Of Presentation:** Very Good
**Impact:** 4

**Recommendation:**

Strong Accept: I recommend accepting the paper and will argue for my recommendation even if other reviewers hold a different opinion.

**Summary:**

The main contribution of this paper is a new large-scale real transparent object dataset including 15000 RGB-D observations that contain RGB, raw depths paired with ground truth depth, instance segmentation, and object pose. The dataset is collected via an automatic procedure using AprilTags, transparent object CAD model, and robot arms. It also introduces a new glass depth estimation method that outperforms the existing approach on both the collected datasets and the existing datasets.

**Issues:**

The main concerning issue is mentioned in the weakness part.

Besides, the writing can be further improved. At least, the notations should be explained when they are used. For example, where does W^p, V^p in Equation (2) come from? It is unclear for me and the authors should not assume they are common notations.

**Reviewer Expertise:**

Very good: Comprehensive knowledge of the area

**Strengths And Weaknesses:**

The presented paper proposed to leverage the distorted depth of the transparent objects from the raw depth, instead of discarding them as did in previous work. The quantitative evaluation shows that this way can achieve higher accuracy.

This paper also presents a new real-world transparent object dataset that can be very useful for the community. It is clearly explained how the dataset was automatically collected using their robotic platform.

One concern regarding the presented dataset is that it only contains 6 objects and a similar background scene (plane). Looking at Figure 3 that shows the CAD model of the used transparent objects, it seems that it actually only contains two different shapes with 3 varying sizes. Considering there are two objects used in the validation and testing and 4 objects in the training, it is a bit worrying if the model is overfitted on the presented datasets. It will be better if the method can also be tested on the shapes in different shapes or on a different dataset to see if it can be generalised well.



**Summary Of Recommendation:**

This paper presented a new method to estimate the depth of transparent objects from an RGB-D image pair. It outperforms the existing approaches. It also presented a new real-world RGB-D transparent object dataset, which will be very useful for the community.

---

> ### Author Response · Authors · 2021-08-25
> **Extended Test Set & Explanation of Notations**
>
> We thank the reviewer for careful review of our work, and the constructive feedback. We provide responses to the points raised as follows:
>
> ***P1: One concern regarding the presented dataset is that it only contains 6 objects and a similar background scene (plane)... It will be better if the method can also be tested on the shapes in different shapes or on a different dataset to see if it can be generalised well.***
>
> A1:  Using the proposed dataset creation pipeline, we collected an extended test set comprising an additional 4000 images for two new objects (bowl and cup), which are significantly different compared to the existing beaker and flask objects in TODD. Overall, TranspareNet’s performance for our extended test set is on par with the result for the original TODD test set. The performance and visualization for our new test set can be found in Appendix A.9.
>
> We provide a copy of Appendix A.9 for reference:
>
> > We introduce an extended test set, consisting of a glass bowl and cup, as shown in Figure 10. The extended test set consists of 4k images of these objects arranged in different settings. We provide an evaluation of TranspareNet when evaluated on the extended test set. Results are shown in Table 6. We further provide visualizations for a representative sample of the new test set, and the quality of the predicted depth in Figure 11.
>
> |                      | RMSE   | MAE    | $\delta$ 1.05 | $\delta$ 1.10 | $\delta$ 1.25 | REL    |
> |----------------------|--------|--------|---------------|---------------|---------------|--------|
> | TranspareNet-DC only | **0.0349** | **0.0308** | 0.3428        | 0.5580        | **0.9228**        | **0.0808** |
> | TranspareNet(ours)   | 0.0363 | 0.0335 | **0.4236**        | **0.5702**        | 0.8417        | 0.0868 |
>
>
> > Our automated dataset creation pipeline allows users to customize the dataset for their intended application scenario by rapidly collecting and annotating images for the transparent objects in the scene. For example, the additional images only took 8 hours to collect and annotate automatically. Although the existing TODD dataset is limited in object types and scenes, we think the pipeline provides a feasible approach to tailor the dataset for specific applications. In Particular, we draw contrast with the method that ClearGrasp used to collect ~200 real-life RGBD images, which involved manual placement, capture, and labelling of each image. Therefore, the automated approach that we used to collect TODD can substantially facilitate future works involving real-life data collection and annotation of transparent objects.
>
>
> ***P2: Besides, the writing can be further improved. At least, the notations should be explained when they are used. For example, where does W^p, V^p in Equation (2) come from? It is unclear for me and the authors should not assume they are common notations.***
>
>
> A2: We have updated the submission with explanations of the notations used.
>
> Thank you for pointing this out. In line 190, we provide an explanation for the notations used for Equation 1.
> > Let $\textbf{m}$ denote the object mask. Given a batch of N inputs with dimension $C^i \times H^i \times W^i$, the output of the SPADE block at site ($n \in N$, $c \in C^i$, $y \in H^i$, $x \in W^i$)
>
> In line 199, we provide an explanation for the notations used for Equation 2.
> > $V$={$v_i$} is collection of all vertices in 3D grid and $W$={$w_i$} is the weights corresponding to each vertex.

---

> > ### Comment · Reviewer_UETv · 2021-09-01
> > **Thanks for solving concern on the test set shapes**
> >
> > Thank you very much for collecting the extra data and providing the results on that.
> > I am now convinced that the proposed approach can be extended to objects that have different shapes to the ones in the training set. I am willing to change the suggestion to the "strong accept".
> > I am still a bit concerned, though, whether the training dataset can cause the approach to overfit it. This is raised by the inconsistency between the TranspareNet-DC only and TranspareNet(ours) on the extended test set. It is also noticed by the performance drop after removing the aprilatg from the environment.
> > Despite this, I still believe the proposed dataset pipeline as well as the method can make a contribution to the conference.

---

> ### Author Response · Authors · 2021-08-30
> **Thanks for your time, any additional feedback is welcome!**
>
> Dear reviewer UETv,
>
> We have addressed *all* the points that you raised with additional clarifications and experiments. Please take a look at our comments, the revised paper and supplemental materials (the latexdiff versions with our changes highlighted are included in the supplementary material for reference). We'd appreciate it if you could let us know if there are any remaining concerns for you.

---

### Official Review · Reviewer_ieyv · 2021-08-11

**Originality:** Very Good
**Technical Quality:** Very Good
**Clarity Of Presentation:** Very Good
**Impact:** 3

**Recommendation:**

Weak Accept: I recommend accepting the paper, but will not argue for my recommendation if the majority of other reviewers have a different opinion.

**Summary:**

This paper contributes a interesting approach and data collection pipeline for what is essentially a shape completion problem (predicting unseen point clouds) that specializes in see-through objects. This is a class of objects that are common and pose a significant challenge to RGB-D sensors we use in robotics. I would say this is an important and relevant problem with a solid approach presented.

**Issues:**

Please see strengths and weaknesses.

**Reviewer Expertise:**

Very good: Comprehensive knowledge of the area

**Strengths And Weaknesses:**

Overall, I am a fan of this paper, good work! The paper addresses an important challenge in perception for manipulation, so props for that. The proposed approach and data-collection pipelines are carefully thought out and well executed. The results do seem quite compelling and I am interested in trying it out. Part of the challenge I feel is the relative recency of the field so benchmarking on actively collected data-sets is still emergent. It is good to see the role that a robot can play in data-collection. I have 2 suggestions to improve the paper:

1- Please provide a detailed discussion on why the work is relevant to manipulation. How good does our estimate of the complete point cloud of the object need to be for various tasks? For example, how well do we need to do shape completion for grasping? Is the existing approach overkill? This sort of discussion of will strengthen the ties of this paper to robotics more.

2- Please provide an evaluation of the normals predicted by the approach -- these have value for the calculation on antipodal grasps.

I also appreciate the acknowledgement of concurrent work as a footnote on the final page of the paper. I think the fact that it is mentioned is sufficient for the initial submission. If it is possible to compare against it for the final submission, that would be excellent.

**Summary Of Recommendation:**

I believe that handling transparent objects is an important step towards general manipulation. I think that this paper is attempting to address this challenge in a convincing way. I do  think this paper should be accepted, it is cool and serves as a good foundation to build upon -- even though personally I don't think the novelty in the network or its losses are that high.

---

> ### Author Response · Authors · 2021-08-25
> **Discussion on Relevance to Manipulation**
>
> We thank the reviewer for the thorough and constructive feedback. Please find our responses to the points raised as follows:
>
> ***P1: Please provide a detailed discussion on why the work is relevant to manipulation. How good does our estimate of the complete point cloud of the object need to be for various tasks? For example, how well do we need to do shape completion for grasping? Is the existing approach overkill?***
>
> A1:
> Our work is directly relevant to perception-based closed-loop manipulation. Most manipulation techniques use point clouds as the perceptual input which is useful for object pose estimation and sim-to-real transfer [A, B, C, D]. Point clouds are deprojected from raw depth maps taken from sensors, so the quality of the depth map directly affects the input for manipulation. In previous works, [E] uses higher quality cameras and [F] observer scenes from multi-viewpoints to deal with sensor noise and occlusion. Recently,  [B, C] take partial point clouds as input and train a multi-stage network on synthetic data. To simulate sensor noise, they employ various data-jittering methods on simulated point clouds. However, these methods assume objects in constructed scenes are opaque and overall geometrical information is preserved in point clouds. For transparent objects, ToF / stereo-based depth sensors cannot accurately capture the depth of transparent and specular objects. Figure  1 in the main text illustrates the quality of the depth estimation by an Intel Realsense Camera where the depth information of the glass beaker is mostly missed or incorrectly captured.
>
> According to ClearGrasp, the effect of an inaccurate depth map on downstream manipulation tasks has been quantified to 12% grasping success rate before any depth estimation methodologies for a parallel jaw gripper, and 64% for suction. This shows the importance of accurate depth estimation for handling transparent objects.
>
> [A] A. Mousavian, C. Eppner, and D. Fox, “6-DOF GraspNet: Variational grasp generation for object manipulation,” International Conference on Computer Vision, 2019
>
> [B] Y. Qin, R. Chen, H. Zhu, M. Song, J. Xu, and H. Su, “S4g: Amodal single-view single-shot se (3) grasp detection in cluttered scenes,” in Conference on robot learning. PMLR, 2020, pp. 53–65
>
> [C] B. Zhao, H. Zhang, X. Lan, H. Wang, Z. Tian, and N. Zheng. REGNet: Region-based grasp network for single-shot grasp detection in point clouds. arXiv preprint arXiv:2002.12647, 2020.
>
> [D] X. Yan, M. Khansari, J. Hsu, Y. Gong, Y. Bai, S. Pirk, and H. Lee. Data-efficient learning for sim-to-real robotic grasping using deep point cloud prediction networks. arXiv:1906.08989, 2019.
>
> [E] J. Mahler, J. Liang, S. Niyaz, M. Laskey, R. Doan, X. Liu, J. A. Ojea, and K. Goldberg. Dex-net 2.0: Deep learning to plan robust grasps with synthetic point clouds and analytic grasp metrics. 2017.
>
> [F]  A.T. Pas, M. Gualtieri, K. Saenko, and R. Platt. Grasp pose detection in point clouds. The International Journal of Robotics Research, 36(13-14):1455–1473, 2017.

---

> ### Author Response · Authors · 2021-08-25
> **Evaluation of Normals**
>
> ***P2: Please provide an evaluation of the normals predicted by the approach -- these have value for the calculation on antipodal grasps.***
>
> We direct the reader to Supplemental Materials -- Appendix A.6 , where we provide a quantitative and qualitative comparison of the surface normals prior to and after depth completion (Figure 7 of the Appendix). Appendix A.6 and the quantitative comparisons are included here for reference; for the qualitative illustrations, please take a look at the Supplemental Materials.
>
> > Appendix A.6:
>
> > In evaluating surface normals, we use the same metrics as ClearGrasp. We compute the mean and median errors of the predicted normal vectors, as compared to ground truth, over all pixels within the image. We also report percentages of the predicted normals which are within 11.25, 22.5, and 30 degrees of that of the ground truth normals. Note that we mask the pixels which include transparent vessels for metric computation.
>
> > Because we do not directly predict normals within our model, we compute surface normals from depth estimations. We consider the depth image to be a function of $z(x,y)$, where $x$ is along the horizontal axis, $y$ is along the vertical axis, and $z$ denotes the depth (in metres).
>
> > The orthogonal vectors tangent to the plane parallel to the $x$ and $y$ axis can then be represented as $(1, 0, dz/dx)$ and $(0,1,dz/dy)$, respectively. Taking the cross product of these vectors, we arrive at the vector which represents the surface normal $(-dz/dx, -dz/dy, 1)$.
>
> > We present an evaluation of the quality of estimated normals based on angular difference with the estimated ground truth normals in Table 3. We also provide a visualization of estimated normals in Figure 7.
>
> |                      |  Mean | Median | $\delta$ 11.25&deg; | $\delta$ 22.5&deg; | $\delta$ 30&deg; |
> |----------------------|:-----:|--------|---------------------|--------------------|------------------|
> | Novel 1 Object Scene | 17.32 |  14.47 |        44.35        |        71.99       |       82.81      |
> | Novel 2 Object Scene | 17.01| 13.92| 46.14| 72.12| 82.62|
> | Novel 3 Object Scene | 19.45| 15.75| 39.14| 65.74| 77.81|
> |Novel Objects Combined |18.57 |15.14 |41.54 |68.17 |79.70|

---

> ### Author Response · Authors · 2021-08-30
> **Thanks for your time, any additional feedback is welcome!**
>
> Dear reviewer ieyv,
>
> We have addressed *all* the points that you raised with additional clarifications and experiments. Please take a look at our comments, the revised paper and supplemental materials (the latexdiff versions with our changes highlighted are included in the supplementary material for reference). We'd appreciate it if you could let us know if there are any remaining concerns for you.

---

### Meta-Review · Area_Chair_Sfqv · 2021-08-11

**Recommendation:** Accept (Oral)
**Confidence:** 4

**Metareview:**

The paper presents an interesting and novel DL approach for an important problem, namely the detection of transparent objects from RGB-D sensors. While on one side, this novelty and the good results of the method are much appreciated by the reviewers, they also raise some important questions regarding the practical applicability of the approach and its tendency to overfit to the comparably small training data. It would be good if the authors could comment on the availabilty of the object masks and how this could be solved in practice, and also explain how the overfitting is addressed. For example, a qualitative evaluation on a data set without AprilTags and with different, random object positions could be considered to show the generalization ability of the approach.

Post-rebuttal:
The discussion between the authors and the reviewers was very helpful. Especially the additional results, showing the usefulness of the approach even without a ground truth segmentation mask and without the AprilTags in the image, have led to a convincing contribution of the paper. The approach is very novel and a good inspiration for the community.

---

> ### Author Response · Authors · 2021-08-25
> **Summary of Responses to Reviews: Solving object mask availability in practice, addressing overfitting**
>
> We would like to thank the reviewers and AC for the thorough and constructive feedback. We are happy to see that all the reviewers appreciate the core idea of our paper for depth estimation.
> * “Interesting approach”, “important and relevant”, “carefully thought out and well executed” (Reviewer ieyv)
> * “Very useful for the community” (Reviewer UETv)
> * “Novel”, “I truly believe this paper deserves a strong accept” (Reviewer vich)
>
> **Summary of responses to the reviews (see individual responses for more detail):**
>
> ***Availability of object masks and how this can be solved in practice***
>
> There are several lines of work which directly address transparent object segmentation given an RGB sensor input, which we mentioned in Related Works - Transparent Object Segmentation. While the presented results in our initial submission used the ground truth segmentation mask, we provide new results in Appendix A.7, which use a segmentation mask pretrained on the Trans10k [15] dataset. These results show that even when using a mask prediction model, and not the ground truth mask, the depth completion quality remains high.
>
> We also provide representative examples of the predicted depths using predicted masks, in comparison to predicted depths using ground truth masks in Figure 8 of the Appendix, showing little difference in the final result and in the estimated surface normals. We believe this shows the applicability and practicality of our model in a real world scenario.
>
> ***Addressing overfitting (qualitative evaluation on dataset without AprilTags)***
>
> We provide an additional ablation study in Appendix A.8, where we generated a mask of the April Tags present in the scene, and masked it out with gray pixels in the RGB stream. We provide a comparison of the inference results using TranspareNet with and without the removal of tags in the RGB stream in Table 5. We also provide a qualitative analysis of the depth estimations for TranspareNet, when the April Tags are masked out in the RGB input in Figure 9 of Appendix.
>
> As can be seen, with and without the presence of April Tags, the model is able to generate very similar depth estimations. This shows that the network indeed learned the geometry of the vessels, and not just the relative positions to the April Tags.
>
>
> ***Addressing overfitting (random objects and positions)***
>
> Using the proposed dataset creation pipeline, we collected an additional 4000 images for two new objects (bowl and cup) which are significantly different compared to the existing beaker and flask objects. The performance and visualization for our new test set can be found in Appendix A.9. Visualizations show that depth inference on the new objects is reliable despite the new geometries.
>
> Additionally, our automated dataset creation pipeline allows users to customize the dataset for their intended application scenario by rapidly collecting and annotating images for the transparent objects in the scene. For example, the additional images only took 8 hours to collect and annotate automatically. Although the existing TODD dataset is limited in object types and scenes, we think the pipeline provides a feasible approach to tailor the dataset for specific applications.

---

> ### Author Response · Authors · 2021-08-30
> **Thanks for your time! We welcome any feedback!**
>
> Dear Area Chair,
>
> We addressed **all** the concerns from the reviewers in our responses to them:
>
> * Reviewer vich: We evaluated the results with a mask segmentation network, conducted an ablation study with April Tags removed from the scene, and updated figures and explanations.
> * Reviewer UETv: We collected an extended test set with shapes drastically different from those in the training set, and added explanations for notations.
> * Reviewer ieyv: We evaluated surface normals, and updated explanations of notations, and layouts of figures, etc.
>
> Since we are nearing the end of the rebuttal period, **we hope that you can encourage Reviewer vich, UETv, ieyv to review the replies and revised paper and supplemental materials** (the latexdiff versions with our changes highlighted are included in the supplementary material for reference) and provide feedback where possible.
>
> We would like to specifically point out that Reviewer vich mentioned that an assessment of the practicality of the network, specifically **using a network to produce segmentation masks is what’s needed for them to recommend a Strong Accept**. See below:
>
> > I hope the authors can clarify the usage of this mask as an input and how they plan to produce it in a real scenario, not just in the constructed dataset test. If they can show that it is also possible to train a mask detection network and use it as an input, with similar or even worse results, then I will change my review to a strong accept.
>
> **We have *fully* addressed this concern** by adapting the Trans10k network, and using the predicted masks to assess the trained models to demonstrate the practicality of our network, without a significant loss in performance. Therefore, we hope that the AC and reviewer can take this into consideration when assessing our work.
>
> At this point, because we have addressed all raised points, we hope to get further justification if the reviewers choose to keep their ratings.

---

### Decision · Program_Chairs · 2021-09-13

**Decision:**

Accept (Oral)

**Comment:**

The paper presents an interesting and novel DL approach for an important problem, namely the detection of transparent objects from RGB-D sensors. While on one side, this novelty and the good results of the method are much appreciated by the reviewers, they also raise some important questions regarding the practical applicability of the approach and its tendency to overfit to the comparably small training data. It would be good if the authors could comment on the availabilty of the object masks and how this could be solved in practice, and also explain how the overfitting is addressed. For example, a qualitative evaluation on a data set without AprilTags and with different, random object positions could be considered to show the generalization ability of the approach.

Post-rebuttal:
The discussion between the authors and the reviewers was very helpful. Especially the additional results, showing the usefulness of the approach even without a ground truth segmentation mask and without the AprilTags in the image, have led to a convincing contribution of the paper. The approach is very novel and a good inspiration for the community.